# The Interconnection between Decent Workplace and Firm Financial Performance through the Mediation of Environmental Sustainability: Lessons from an Emerging Economy

Muhammad Zahid [1,2,*], José Moleiro Martins [3,4], Haseeb Ur Rahman [5,*], Mário Nuno Mata [3,6], Syed Asim Shah [7] and Pedro Neves Mata [8]

1 Department of Management Sciences, City University of Science and IT, Peshawar 25000, Pakistan
2 City University Center for Sustainability Studies (CUCSS), Peshawar 25000, Pakistan
3 ISCAL (Instituto Superior de Contabilidade e Administração de Lisboa), Instituto Politécnico de Lisboa, Avenida Miguel Bombarda, 20, 1069-035 Lisboa, Portugal; zdmmartins@gmail.com (J.M.M.); mnmata@iscal.ipl.pt (M.N.M.)
4 Business Research Unit (BRU-IUL), Instituto Universitário de Lisboa (ISCTE-IUL), 1649-026 Lisboa, Portugal
5 Institute of Management Sciences, University of Science and Technology, Bannu 28100, Pakistan
6 Polytechnic Institute of Santarém, School of Management and Technology (ESGTS-IPS), 2001-904 Santarém, Portugal
7 Department of Management Sciences, Attock Campus, COMSATS University Islamabad, Punjab 43600, Pakistan; sashah006@gmail.com
8 ISTA-School of Technologies and Architecture, Instituto Universitário de Lisboa (ISCTE-IUL), ISTAR-IUL, Avenida das Forças Armadas, 1649-026 Lisboa, Portugal; pedronmata@gmail.com
* Correspondence: zahid@cusit.edu.pk (M.Z.); haseebbabo@gmail.com (H.U.R.)

**Abstract:** This study aimed to investigate the impact of some important Sustainable Development Goals (SDGs), such as the decent workplace, climate change, and economic sustainability on firm financial performance (see Goals 8 and 13). By adopting an index from the previous literature, this study collected data from the annual and sustainability reports of the publicly listed companies of a developing country through content analysis from 2016 to 2018. The results revealed a significant increase in the level of compliance with workplace and environmental sustainability during the corresponding period. Furthermore, the estimations of ordinary least squares (OLS) and two-stage least squares (2SLS) panel data also unveiled a positive impact of workplace sustainability on the firm's environmental and financial performance. Additionally, we noted that the findings were pronounced after addressing the problem of endogeneity. Moreover, the study also found a novel significant and positive mediating role of environmental sustainability in the relationship between workplace sustainability and the firm's financial performance. This study has theoretical significance by proposing sustainability training and development as instrumental variables in the relationship of the workplace and environmental sustainability to firm financial performance. This study offers practical implications for regulatory bodies and business firms to integrate workplace and environmental sustainability practices into their routine operations for achieving sustainable industrialization.

**Keywords:** decent workplace; sustainability practices; financial performance; index; endogeneity

## 1. Introduction

Achieving workplace sustainability has become one of the key topics around the world, especially in developed countries, where people are more concerned about the workplace environment. The awareness regarding sustainable workplaces has also increased in developing countries, as the governments of some of these countries have established their long-term development plans, such as adhering to the Sustainable Development Goals (SDGs) [1,2]. Likewise, the importance of sustainability at the corporate level has also been

significantly increased, especially after the introduction of SDGs by the United Nations Development Program (UNDP) in 2017 [3]. All the SDGs are inter-related; however, some SDGs are the focus of this study, such as SDG 8, which promotes sustained, inclusive, and sustainable economic growth, full and productive employment, and decent work for all. Similarly, SDG 13 advocates taking urgent action to combat climate change and its impacts on the broader environment (for further detail and targets, see SDGs 8 and 13) [3]. The term "sustainable development" is defined as "development which meets the needs of the present without compromising the ability of future generations to meet their own needs" [4]. To address sustainability in the workplace, the SDGs underline decent work, climate change (environmental sustainability), and economic growth (financial performance) in the aim of achieving full and productive employment, in addition to decent work, by the end of 2030. An important dimension of decent work is workplace sustainability, which is concerned with work and the work-related issues of organizational employees. Workplace sustainability addresses the human rights, health, safety, working conditions, training and development, and social issues of the employees, in addition to their relationship with the employer [5,6]. The authors refer to workplace sustainability as a practice of providing such an environment for work that has a direct positive impact on employees' performance, productivity, and motivation [7].

In today's corporate world, sustainability is considered a highly important issue, and corporations guarantee the fortification of different stakeholders by safeguarding their interests through corporate sustainability practices [8]. Hence, it has become one of the most critical agenda of these organizations, especially in making strategic decisions. Additionally, sustainability-related issues have caught the attention of policymakers and regulators to be included in the governance mechanism and duties of the board of directors [9]. The descriptive approach of the stakeholder theory also postulates that the management and employees of the firm have the responsibility of managing the interests of a broad spectrum of stakeholders [10]. Considering the importance of sustainability in the workplace, it is believed to be a key driver for the overall sustainability of an organization. The sustainable workplace augments and promotes firms' overall sustainable conduct [11,12]. However, despite a plethora of publications about the sustainable workplace and sustainability reporting of corporations [13], the nature of research in the area mostly remains exploratory or conceptual. A quantitative examination of a sustainable workplace has been greatly overlooked in prior research. The area of the sustainable workplace, which is also known as green human resource management, is an emerging phase. The most recent works of authors have tried to develop a proper instrument to measure workplace sustainability [14,15]. Similarly, the center of focus of previous studies was to judge the perception of employees and top management with regard to the implementation of workplace sustainability within an organization's operations [16,17]. However, only a few of these studies could focus on workplace sustainability practices by employing the content analysis method of data collection from the annual reports of the firms [18]. To fill this evident gap, the current study formulates the following research questions: how do workplace sustainability practices impact firms' environmental and financial performances? Additionally, what is the mediating role of environmental sustainability in the relationship between workplace sustainability and firm financial performance?

To answer these questions, this study selected a developing country, such as Malaysia, as the research context. The context we selected is significant, as it is one of the fast-growing economies in the Asian region, which promulgated several regulatory steps to implement sustainability in its business firms, including health and safety, workplace sustainability, and mitigating negative environmental impacts, as well as achieving sustainable growth and industrialization. Additionally, this context is further important, as sustainability reporting in developing and emerging economies is significantly lower compared to that of developed countries. The firms in the selected context face a large number of workplace-related accidents, and these are increasing across all sectors of the economy, as per the Report of the Department of Occupation Safety and Health (DOSH) [19]. This might be

due to the fact that firms in the country, as in other developing countries, mainly focus on external stakeholders; they remain engaged in philanthropic activities and ignore the wellbeing of their employees. This triggers the need for managerial attention to improve the working environment and ensure workplace sustainability. Amongst others, one of the mitigating strategies to reduce the number of these accidents and hazards could be the adoption of workplace and environmental sustainability.

By answering the research questions, this study would have many contributions. First, unlike previous researchers, the current study offers a theoretical contribution for investigating actual workplace and environmental sustainability practices, rather than the perceptual aspects of the stakeholders [16,17]. Second, the study applied the crux of stakeholder theory about managing multiple stakeholders, including internal and external. Third, the study proposes a novel index to measure workplace and environmental sustainability practices. Fourth, the study brings methodological contributions by adopting longitudinal data for investigation. Additionally, the study controls the problem of endogeneity by introducing a unique instrumental variable (sustainability training and development) in using 2SLS instrumental variable approaches, which might be one of the reasons for inconclusive findings in the past [20,21]. Furthermore, the study also tests the novel mediation of environmental sustainability in the relationship between workplace sustainability and financial performance. Finally, the study has social and practical implications for developing and emerging economies to implement the agenda of the decent workplace, mitigate climate change, and achieve economic sustainability by attaining the overall objectives of cleaner production, as well as sustainable industrialization.

The remaining sections of the study are structured as follows: the next part deals with the literature review, which is followed by research methods, results, and discussions. The last section consists of a conclusion and future recommendations for future studies.

## 2. Literature Review

### 2.1. Workplace Sustainability

The importance of sustainable development has increased significantly in business firms after the promulgation of the Sustainable Development Goals (SDGs) by the United Nations Development Program (UNDP) in 2017. Among these SDGs, some are directly related to business firms, such as decent workplace and climate change initiatives (see Goal 8 and 13). Similarly, corporate sustainability is a subset of the sustainable development introduced in the Brundtland report in 1987, with the vision to balance the environmental, social, and economic sustainability of human civilization [22]. Firms pursuing these in their operations name it as corporate sustainability and reporting. The Global Reporting Initiative (GRI) introduced a corporate sustainability reporting framework [23] that covers environmental, social, and economic dimensions. The internal (employees, workers, and workplace-related issues) and external stakeholders (community and society) are the subjects addressed under social sustainability. Employees, being important internal stakeholders, are supposed to be fairly treated by firms [24]. Under workplace sustainability, the GRI requires firms to encourage workforce diversity and equal employment opportunities, irrespective of gender or skin color, for retaining employees, increasing firms' reputations, and reducing risks [25,26]. Furthermore, freedom of association and collective bargaining are also inclusive, as the absence of these or independent labor unions may affect the workplace [27]. A corruption-free environment [28], decent labor practices, shelters for employees [29], discouraging child, forced and compulsory labor, as well as protection of human rights are among the agendas of workplace sustainability [30]. Workplace sustainability also requires firms to address the issues related to the health, safety, and welfare of employees. The Occupational Safety and Health Administration Act (2016) explains that workplace sustainability deals with the triple bottom line—planet, people, and profit—to gain success in the long-term. The Act explains that these measures assist firms in gaining the loyalty, commitment, and motivation of employees. Besides, workplace sustainability also focuses on ethical responsibilities, as well as protection or

improvement of the natural environment [1]. In addition to employees' health and safety, the environment is also an essential consideration of workplace sustainability. The GRI focuses on employees' education, trainings, skills' development and welfare for developing a sustainable workplace [31].

*2.2. Theoretical Framework and Hypotheses Development*

2.2.1. Workplace and Environmental Sustainability

The stakeholder theory postulates that organizations have obligations not only to shareholders but to the other groups of stakeholders, such as customers, suppliers, employees, and the wider community, amongst many others [10,24]. Meeting the demand of stakeholders is necessary for an organization to sustain and continue the supply of resources, and for legitimation reasons [32]. The stakeholder theory treats employees as internal stakeholders of the company, and their fair treatment enhances firms' environmental and financial performances [18,24]. Workplace sustainability establishes a bond between firms and employees by addressing workplace-related issues [8,31]. Among others, this improves or at least neutralizes the possible non-favorable behavior of employees towards firms [33]. It is noted that workplace sustainability awareness, individuals' knowledge, infrastructure, and collaborations may boost environmental sustainability at the corporate level [34]. The diversion of firms' attention towards workplace sustainability or, more specifically, to the green attitudes and behaviors of employees and strategies, assist them in achieving their financial, social, and ecological goals [35]. Workplace sustainability keeps focusing on employees for enhancing firms' environmental sustainability [36]. Firms may acquire or enrich their green abilities by the self-efficacy gained through recruitment and selection of new skilled employees or by training already existing employees who could boost their environmental outcomes. They assist firms in promoting cleaner and environmentally friendly production, recycling, and controlling or reducing emissions [37]. Workplace sustainability is considered one of the most important and significant policies of human resource management (HRM) to increase the green spending of resources in organizations and to encourage the cause of a sustainable environment [38]. Similarly, workplace sustainability maintains and improves knowledge capital through the implementation of human resource (HR) policies and activities that are environmentally friendly. Hence, it is hypothesized that:

**Hypothesis 1 (H1).** *Workplace sustainability will be positively associated with firm environmental sustainability.*

2.2.2. Environmental Sustainability and Financial Performance

The questions asked are whether firms go green to get any reward or whether they do so to improve their financial performances [39]. Many studies, including a meta-analysis, say "Yes" to these questions, and show a positive link between environmental sustainability and firm financial performance [40,41]. However, some studies also noted a negative relationship between environmental sustainability and firms' financial performances [42,43]. These studies explain that environmental sustainability has a higher cost that affects firms' subsequent financial performances. Additionally, they argue that environmentally friendly production requires the latest, sophisticated, and cleaner technologies that have higher costs with no immediate return [44]. Some of the studies also found no significant relationship between environmental sustainability and firms' financial performances [45,46]. These studies concluded that environmental sustainability is unprofitable and inappropriate for firms [47,48]. However, as a solution, some studies suggested that firms may proactively implement environmental sustainability plans and respect the environment into their routine activities by considering and reporting its high cost as a capital expenditure. Hence, it is hypothesized that:

**Hypothesis 2 (H2).** *Environmental sustainability will be positively associated with firm financial performance.*

2.2.3. Workplace Sustainability and Financial Performance

By following workplace sustainability, firms satisfy employees, enhance their productivity, and, thus, improve their financial performance [49]. Working conditions account for dignity, equality, and social protection of individuals, or employees increase productivity, motivation, and reputational capital [50]. Reputational capital contributes towards job satisfaction and a lower turnover of employees due to firms' positive image in the minds of employees, their families, relatives, and friends [51]. These improve firms' financial performances as high morale and motivation of employees increase their productivity [44]. Among others, workplace sustainability and its reporting are also imperative to improving performance, long-term economic benefits, and the competitive advantage of firms [52]. By improving sustainability at the workplace, or, more specifically, increasing human dignity, equality, and social protection [50], firms improve productivity and overall performance through satisfying and motivating their employees [49]. It also assists firms in gaining ethical and reputational benefits, which, among others, decrease employee turnover. Furthermore, workplace sustainability also encourages the training and development of employees, decentralization, and participative leadership style (Berman et al., 1999). These, in turn, improve firm performance [49]. The authors [53] noted that small and medium enterprises (SMEs) achieve superior financial performance than their competitors by improving workplace sustainability. They further noted that improving workplace sustainability progresses employees' productivity, products' quality, and firms' financial performances. Hence, it is hypothesized that:

**Hypothesis 3 (H3).** *Workplace sustainability will be positively associated with firm financial performance.*

2.2.4. Workplace, Environmental Sustainability, and Financial Performance

Workplace sustainability integrates firms' environmental objectives in human resource functions such as recruitment, training, and reviewing the performance and remuneration of employees to assist firms in ensuring carbon emission and earning carbon credits [54]. Earlier, firms were supposed to have better financial performance; however, now it must be accompanied by care for society and the environment. The corporate environmentalism or green management that emerged in the 1990s required firms to develop environmental management strategies and ensure an industrial growth, along with the protection of the environment for future generations [15]. It refers to firms' interaction with the environment for controlling pollution and improving product stewardship and corporate social responsibility. In doing so, firms are encouraged by the use of the latest innovative technologies to reduce environmental deterioration and develop biotech products. Firms are also supposed to ensure the use of alternative energies to decrease pressure on specific natural resources. They should focus on research, innovation, and technology to develop nontoxic products and protect the environment. Studies noted that workplace sustainability preserves resources and decrease firms' negative impacts on the environment and its inhabitants [1]. These, in turn, affect the outlook or perception of firms, which improves their financial performances by influencing the buying behavior of customers and the responses of shareholders and investors [52]. The author further endorsed that developing a sustainable workplace enhances firms' productivity and financial performances. Hence, it is hypothesized that:

**Hypothesis 4 (H4).** *Environmental sustainability will mediate the relationship between workplace sustainability and firm financial performance.*

2.2.5. Conceptual Framework

The conceptual framework of the study is based on stakeholder theory. Figure 1 explains the flow of hypotheses, the straight arrows present direct relationships while the dotted arrow presents indirect (mediating) relationships from workplace sustainability to firm financial performance.

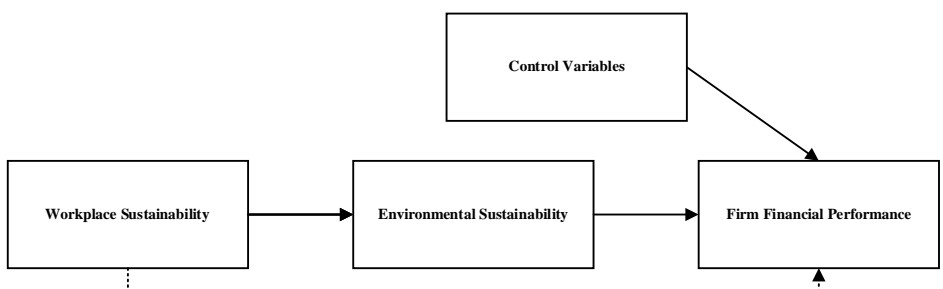

**Figure 1.** Conceptual framework.

## 3. Research Methods

### 3.1. Population and Sample

This study used Malaysian public listed companies as a unit of analysis. Malaysia is one of the fast-growing emerging economies and committed to sustainable development [55]. In pursuing this, the country has already started promoting sustainability in public listed companies, as evidence in and after the Eleventh Malaysian Plan. The Malaysian stock exchange (the Bursa Malaysia), introduced four dimensions of sustainability, including community, workplace, environment, and marketplace; where the community deals with society and external stakeholders, while the workplace accounts for internal stakeholders, the environment, and the economic contributions of firms. Being the focus of the current setting, Bursa Malaysia explained workplace sustainability as firms' social responsibility regarding human rights, gender equality, working conditions, health and safety of the employees, environment, and society [56]. As a whole, Bursa Malaysia assumes sustainability as ethical and transparent business practices that contribute to the community, employees, environment, shareholders, stakeholders, and society at large [18,57]. The evaluation of prior literature shows that Malaysian firms follow and report their environmental and community-related sustainability practices. However, they pay less attention to workplace sustainability or its reporting [58]. These studies noted that sustainability, and especially that related to the workplace, is still low and cosmetic, even after passing a decade of the introduction of Bursa Malaysia guidelines [57]. A few recent studies found some improvements in compliance and reporting to workplace sustainability [18]. Hence, the new investigation may add new insights into the area. Hence, to look at workplace sustainability in Malaysian public listed companies, the study used 900 reports (300 reports per year X 3) from 12 sectors for 3 years from the period 2016 to 2018. During the period the Malaysian government and Bursa Malaysia, in particular, took several steps, such as the introduction of the Malaysian Code of Corporate Governance (MCCG) 2017, introducing several sustainability-related awards, issuing of a director's guide for the implementation of sustainability practices, and stressing the adoption of GRI standards to implement sustainability practices, with the true spirit in the public listed companies of Malaysia [8]. The study preferred stratified random sampling and the companies were selected from each section using a random sampling technique. Additionally, from the sectors with a small population, i.e., less than 10, all the companies/reports were included. The breakdown of the sample used for the study is shown in Table 1.

**Table 1.** Sector-wise Sample Companies.

| No. | Sectors | No. of Sample Companies | Percent |
|---|---|---|---|
| 1. | Consumer | 30 | 10 |
| 2. | Trading | 51 | 17 |
| 3. | Industrial | 48 | 16 |
| 4. | Plantation | 24 | 8 |
| 5. | Hotels | 6 | 2 |
| 6. | Real estate | 15 | 5 |
| 7. | Infrastructure | 9 | 3 |
| 8. | Properties | 30 | 10 |
| 9. | Technology | 33 | 11 |
| 10. | Finance | 9 | 3 |
| 11. | Construction | 42 | 14 |
| 12. | Mining | 3 | 1 |
| | Total | 300 | 100 |

The data on workplace and environmental sustainability practices were collected from the company annual reports through a content analysis approach using the data collection index (see Appendix A) adapted from the previous authors [59,60]. The authors advocate that content analysis is the most widely used approach in research, especially for extracting quantitative data from annual reports. To record each of the contents, the study utilized the binary coding of 0 and 1 that assigns 1 if the company reports the content, and 0 otherwise [61,62]. Simply, the highest score for using workplace and environmental sustainability = $\Sigma di\ n$ to sum the contents, which means the high performance of a company and vice versa [59,60]. On the other hand, financial data, including control variables, were also collected from the annual reports of sample companies. Financial performance was measured in terms of return on equity (net income/shareholder equity) (ROE) [8,21]. The study also controls the firm size (log of the company total assets) [21], firm age (the time duration in years since its incorporation till the time of observations) [8,63], leverage of the firm (total liabilities/total shareholders' equity) [8], and industry type and years (Industry and year as dummy variables representing the industry sector and year) [60,64].

*3.2. Statistical Techniques*

For the hypotheses testing, the study applied pooled ordinary least squares (OLS) and two-stage least squares (2SLS) statistical techniques, as recommended by the previous authors. The authors further argued that sustainability data might be endogenous due to omitted variables bias, measurement error and reverse causality [21,65,66]. To identify such errors, the study applied an ovtest in Stata 13. The test reveals that, in all the models, there was a problem of endogeneity and, hence, addressed this problem by using a proper instrumental variable in 2SLS.

The study assumed and applied corporate sustainability training as an instrumental variable. The rationale behind this was to increase contributions towards sustainability; firms must establish human resource management practices that support their desired strategy and core values [67]. Sustainability training and development provide a proper mandate to employees and a source of sustainability culture within a firm. Besides, it is also a source of firms' transformation to sustainable best practices. By conducting such training and development programs, firms will benefit by increasing compliance with regulatory requirements, understanding the satisfaction of stakeholder demands, organizational and employee responsibilities, gaining a positive public image, employee motivation and better productivity, and firm financial performance [68]. Training and development for sustainability integration is an important factor for the optimum utilization of firms' resources and efforts [69].

To refine the findings further, for the relationship and the control of the problem of reverse causality, the study applied a one year-lag of dependent variables (environmental

sustainability and firm financial performance) in the corresponding models. Moreover, the lag of the dependent variable also controlled the problem of autocorrelation in the data [65]. After applying instrumental variables in 2SLS, the p-value of Durbin and Wu-Hausman (1978) tests above 0.05 provided evidence that the endogeneity problem had been addressed [21]. Furthermore, the Breusch-Pagan/CookeWeisberg and imtest were applied to identify the problem of heteroscedasticity. The tests revealed that some models were violating the problem of homoscedasticity and, hence, correctly applied the robust standard errors in all the models [65,70]. The following econometric models applied in OLS and 2SLS estimations.

$$\gamma_{it} = a1 + \beta 1_{it} + \gamma 1 x_{it} + \delta_t + ni + \varepsilon_1 it \tag{1}$$

$$\beta 1_{it} = a2 + \theta Z_{it} + \gamma 2 X_{it} + \delta_t + ni + \varepsilon_2 it \tag{2}$$

where:

$\gamma$ = Dependent Variable(s)
$it = i$ Representing for Firm and $t$ Representing Year
$\alpha_1, \alpha_2$ = Constants of 1st and 2nd stage, respectively
$\beta_{1it}$ = Endogenous Independent Variable(s)
$\gamma_1 \chi_{it}, \gamma_2 \chi_{it}$ = Control Variables in 1st and 2nd stage, respectively
$\delta_t$ = Year Dummies
$\eta_i$ = Industry Dummies
$\theta Z_{it}$ = Instrumental Variable
$\varepsilon_{1it}, \varepsilon_{2it}$ = Error Terms of 1st and 2nd stage, respectively

## 4. Results and Discussion

Table 2 exhibits descriptive statistics for ROE, workplace and environmental sustainability, firm size, age, and leverage of the sample companies from 12 different sectors. The descriptive statistics report means, maximum, minimum, standard deviation, skewness, and kurtosis values of the variables. ROE reports a minimum value of –1.113, maximum of 0.89, and mean value of 0.09. Workplace reveals a minimum of 1, maximum of 17, and a mean value of 5.83. Similarly, environmental sustainability records a minimum value of 2 and a maximum 12, while the mean value is 4.43. Firm size reports 4.440 minimum, 9.53 maximum, and 5.569 of mean values. Firm age recorded 1 as minimum, 43 as maximum, and 15.99 as mean values of the ages of the companies. Finally, the firm leverage reports a 0.000 minimum, a 67.87 maximum, and 1.37 as the mean value. All the skewness and kurtosis values are reported based on normalized values using the Van der Waerden transformation method.

**Table 2.** Descriptive Statistics.

|  | Min | Max | Mean | S.D | Skewness | | Kurtosis | |
| --- | --- | --- | --- | --- | --- | --- | --- | --- |
|  |  |  |  |  | Stat | S.E | Stat | S.E |
| Return on equity | –1.113 | 0.89 | 0.090 | 16.549 | 0.000 | 0.083 | –0.098 | 0.165 |
| Workplace sustainability | 1 | 17 | 5.83 | 2.645 | 1.234 | 0.083 | 2.222 | 0.165 |
| Environmental sustainability | 2 | 12 | 4.43 | 2.865 | 0.492 | 0.083 | –0.425 | 0.165 |
| Firm size | 4.440 | 9.53 | 5.569 | 0.755 | 0.029 | 0.083 | –0.272 | 0.165 |
| Firm age | 1 | 43 | 15.99 | 8.036 | 0.134 | 0.083 | –0.661 | 0.165 |
| Firm leverage | 0.00 | 67.87 | 1.37 | 0.216 | –0.029 | 0.083 | –0.155 | 0.165 |

Table 3 reports the statistics for the Pearson correlation matrix. None of the correlations between two predictors is higher than 0.8, and, hence, there is no multicollinearity in any association [65].

**Table 3.** Pearson's correlation matrix.

| | (1) | (2) | (3) | (4) | (5) | (6) |
|---|---|---|---|---|---|---|
| Return on equity (1) | 1 | | | | | |
| Workplace sustainability (2) | 0.187 ** | 1 | | | | |
| Environmental sustainability (3) | 0.084 * | 0.630** | 1 | | | |
| Firm size (4) | 0.153 ** | 0.300 ** | 0.231 ** | 1 | | |
| Firm age (5) | –0.083 * | 0.166 ** | 0.147 ** | 0.189 ** | 1 | |
| Firm leverage (6) | –0.015 | 0.085 * | 0.034 | 0.234 ** | 0.005 | 1 |

** Correlation is significant at the 0.01 level (2-tailed). * Correlation is significant at the 0.05 level (2-tailed).

Table 4 reports that there is a positive impact of workplace sustainability on firm environmental sustainability practices in both OLS and 2SLS estimations. Moreover, the results of 2SLS are more pronounced than the OLS after addressing the problem of endogeneity. The findings support H1 of the study and explain that workplace sustainability has importance for the environmental sustainability of a firm. These findings endorsed the postulation of stakeholder theory that firms are responsible for satisfying the demands of multiple stakeholders within and outside the organization. Our findings are consistent with many previous studies [33,37].

**Table 4.** Regression results.

| | (OLS) Environmental | (2SLS) Environmental | (OLS) ROE | (2SLS) ROE | (OLS) ROE | (2SLS) ROE |
|---|---|---|---|---|---|---|
| Workplace sustainability | 0.632 *** | 0.839 *** | | | 0.871 *** | 1.267 ** |
| | (0.030) | (0.075) | | | (0.204) | (0.541) |
| Environmental sustainability | | | 0.427 ** | 2.028 *** | | |
| | | | (0.194) | (0.648) | | |
| Firm size | 0.047 *** | 0.033 *** | 0.231 *** | 0.117 | 0.212 ** | 0.155 |
| | (0.012) | (0.013) | (0.083) | (0.112) | (0.090) | (0.096) |
| Firm age | 0.009 | 0.005 | –0.248 *** | –0.348 *** | –0.257 *** | –0.371 *** |
| | (0.010) | (0.010) | (0.070) | (0.076) | (0.068) | (0.072) |
| Firm leverage | 0.000 | –0.000 | –0.005 ** | –0.005 * | –0.005 * | –0.005 * |
| | (0.000) | (0.000) | (0.002) | (0.003) | (0.003) | (0.003) |
| Lag of DV | –0.009 | –0.010 | 0.335 *** | –0.018 | 0.327 *** | 0.685 *** |
| | (0.025) | (0.026) | (0.031) | (0.190) | (0.058) | (0.116) |
| Constant | –1.479 | –2.493 * | –13.431 ** | 2.351 | –15.314 ** | –4.950 |
| | (0.941) | (1.446) | (6.479) | (11.734) | (6.021) | (12.419) |
| Obs. | 877 | 877 | 877 | 877 | 877 | 877 |
| R-squared | 0.457 | 0.427 | 0.202 | 0.025 | 0.214 | 0.161 |
| Durbin (score) chi2(1) | | 9.754 | | 7.678 | | 1.415 |
| $p$–value | | 0.065 | | 0.064 | | 0.234 |
| Wu-Hausman F | | 9.639 | | 7.570 | | 1.385 |
| $p$–value | | 0.075 | | 0.061 | | 0.239 |
| Years Dummy | Yes | Yes | Yes | Yes | Yes | Yes |
| Industry Dummy | Yes | Yes | Yes | Yes | Yes | Yes |

Standard errors are in parenthesis. *** $p < 0.01$, ** $p < 0.05$, * $p < 0.1$.

The statistics of OLS and 2SLS shown in Table 4 further reveal that there is a positive impact of the workplace and environmental sustainability on firm financial performance using ROE as a proxy. Similarly, the results are more noticeable in the 2SLS estimator than OLS after addressing the problem of endogeneity.

The findings support H2 and H3 of the study, respectively, and endorsed that there is a positive impact of the workplace and environmental sustainability on firm financial performance using ROE as a proxy. The findings are in line with the explanation of stakeholder theory and many other studies which found that a firm's focus on workplace

and environmental sustainability provides instrumental benefits in the shape of better financial performance [1,44,49].

For testing indirect effect, the study applied a Stata sgmediation command. Unlike the conventional four-steps Baron and Kenny (1986) approach, the current study applies the significance of an indirect effect for a valid mediation, as suggested by the previous literature [69–71]. The results of the sgmediation command of Stata 13 reported in Table 5 showed that there is a significant positive mediation between the relationship of the workplace and environmental sustainability practices. The findings support H4 of the study and implied that workplace sustainability improves firms' environmental sustainability and, hence, provides instrumental benefits in the shape of better financial performance. The results are in line with the postulation of stakeholder theory and previous studies [54].

**Table 5.** Mediation analysis.

| Mediation Model | | | | |
| --- | --- | --- | --- | --- |
| Independent Variable | Mediating Variable | Dependent var. | Indirect Effect (*t*–Value)/ Confidence Interval | Mediation |
| Workplace Sustainability | Environmental Sustainability | Financial Performance (ROE) | 3.97 [0.121–0.328] | Yes |

Note: The results of the indirect effect and confidence interval are based on 5000 bootstrap tests.

## 5. Conclusions, Limitation and Future Directions

This study investigated the impact of workplace sustainability practices on firm environmental sustainability and financial performance in the context of a developing country. The unique findings of the study reveal that workplace sustainability has a significant positive impact on environmental sustainability. Additionally, the findings explain that workplace sustainability boosts green attitudes, behaviors, and strategies of the employees, and, hence, the firm achieves environmental sustainability. Such initiatives may help a firm to enrich their green abilities by the self-efficacy gained through the recruitment and selection of new skilled employees or the training of already existing employees who could boost their environmental outcomes. Besides, the positive impact of environmental sustainability on a firm's financial performance endorsed the stakeholder theory that a firm focusing on the environment may reap better financial performance. The findings also oppose the notion that the high cost of environmental practices negatively affects a firm's financial performance. The findings for novel mediation endorse a sequential link among the workplace, environmental sustainability, and firm financial performance. The findings endorsed that firms that are involved in workplace sustainability practices may improve employees' productivity, products' quality, environmental behaviors, firms' cash flow, and overall financial performance. By doing so, firms may satisfy internal and external stakeholders, which may create reputational benefits, contribute to high job satisfaction, and lower employee turnover. As a result, it increases firms' financial performance, and high morale and motivation of employees that increase their productivity.

The study has many contributions. First, this study used the actual workplace and environmental sustainability practices instead of the perceptual aspects of the stakeholders, as used by most of the prior studies [16,17]. Second, the study adopted a comprehensive approach by considering both the internal and external stakeholders, as explained by the stakeholder theory. Third, the study offered a unique index for gauging workplace and environmental sustainability. Fourth, unlike most of the prior literature, which relied on cross-sectional methodology, this study used longitudinal data, which further adds to the clarity of the relationships. Besides, the use of 2SLS for proposing novel instrumental variables (sustainability training and development), in using 2SLS instrumental variable approaches, are also the methodological contributions of the study [20]. In addition to the literature and methodology, the study also has a fifth contribution by providing empirical evidence to support the sequential process of stakeholder theory by proposing and investigating the novel mediation of environmental sustainability of the relationship

between workplace sustainability and financial performance. Finally, the findings motivate developing and emerging economies to ensure the implementation and development of the SDGs-Goals, such as the decent workplace and the mitigation of climate change, to achieve cleaner and sustainable forms of industrialization.

The current study is not free from the limitations. First, the results of the current work are based on three years of data and, hence, in the future, the dataset may be extended for further enriching the findings. Second, the study is based on the overall analysis of the industry; separate industrial analysis, dividing the industry into environmentally friendly and non-friendly for the workplace, and environmental sustainability practices may add worth to the scholarship in the future. Third, future studies may consider corporate governance variables in the relationships between workplace, environmental, and financial firm performances. Fourthly, some other developing countries should also be considered in future studies for the generalizability of the results. Last but not least, the aspects of qualitative and mixed methods may also be considered for further refinement of the findings, or alternative explanations in the future.

**Author Contributions:** Conceptualization, M.Z., H.U.R.; data curation, S.A.S.; formal analysis, M.Z.; funding acquisition, J.M.M., P.N.M.; resources, M.N.M. All authors have read and agreed to the published version of the manuscript.

**Funding:** This research was supported by Instituto Politécnico de Lisboa.

**Institutional Review Board Statement:** Not applicable.

**Informed Consent Statement:** Not applicable.

**Data Availability Statement:** No new data were created or analyzed in this study. Data sharing is not applicable to this article.

**Conflicts of Interest:** The authors declare no conflict of interest.

## Appendix A

**Table A1.** Workplace and Environmental Sustainability Disclosures Index.

| Workplace Sustainability | Environmental Sustainability |
| --- | --- |
| Decent Labor Practices | Environmental Management System (EMS) and Certifications |
| Minimum Wages for employees | Material Used and Produced |
| Workplace Ethical Values | Material Recycled |
| Employment Opportunities | Energy Consumption and Reduction |
| Occupational Health and Safety | Water Consumption |
| Training and Development | Biodiversity |
| Diversity and Equal Opportunities | Emissions including Greenhouse Gases (GHG) |
| Supplier's Labor Assessment | Effluents and Waste Reductions |
| Protection of Human Rights | Product Environmental Impacts |
| Collective Bargaining Power | Transportation Impacts |
| Prevent Child and Compulsory Labor | Suppliers' Environmental Impacts |
| Employees Satisfaction Survey | Environmental Related Awards |
| Shelters for Employees and their Family | |
| Anti-Corruption | |
| Sports and Work–life Balance | |
| Workplace Related Awards | |

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
