# Peer review of "The Interconnection between Decent Workplace and Firm Financial Performance through the Mediation of Environmental Sustainability: Lessons from an Emerging Economy"

_sustainability, doi:10.3390/su13084570_

Round 1
Reviewer 1 Report
The theme of the study is topical. Coherently with the greater relevance given to the SDGs, it is very appreciated the choice of this theme. It is interesting the investigation of the relationship between workplace sustainability with Malaysian firms’ financial performance through the mediating effect of the environmental sustainability.
The abstract could be summarized in less text.
The introduction is well organized. The aim of the study, the research gap to fill, the research question, the contributions and the implications are explained in a clearly manner.
The literature review section sheds light about the motivations under the choice of the Malaysian context. The theoretical framework and hypothesis development are well argued. However, it could be specified the sustainable development goal to which sustainability workforce refers.
The conceptual framework (figure 1) is clear, especially in the representation of the indirect/mediating relationship between workplace sustainability and firm financial performance through the environmental performance.
The research methods section could be reorganized. The section should embrace the statistical techniques (OLS and 2SLS) involved to test the hypothesis. This aspect is included in the Results and Discussion section, but it is not the appropriate section.
The construction of sustainable workforce and environmental sustainability variables is not clear (lines 277-281).
I suggest to clarify the type of reports from which data were extracted, how the content analysis is developed and the software used to do it. The sources of the financial and control variables should be known.
The time span adopted (2016-2018), based on three years’ data, is not very wide to verify the effects of the implementation of the sustainable workplace policies on the environmental sustainability and financial performance.
With reference to the limitations of the study, it could be considered that the results cannot be generalized over the Malaysian context.
Author Response
The abstract could be summarized in less text.
Author response: The abstract is now reduced accordingly.
The introduction is well organized. The aim of the study, the research gap to fill, the research question, the contributions and the implications are explained in a clearly manner.
Author response: N.A
The literature review section sheds light about the motivations under the choice of the Malaysian context. The theoretical framework and hypothesis development are well argued. However, it could be specified the sustainable development goal to which sustainability workforce refers.
Author response: Now it is linked with the SDG No. 8, 13, and 17 in the introduction section.
The conceptual framework (figure 1) is clear, especially in the representation of the indirect/mediating relationship between workplace sustainability and firm financial performance through environmental performance.
Author response: N.A
The research methods section could be reorganized. The section should embrace the statistical techniques (OLS and 2SLS) involved to test the hypothesis. This aspect is included in the Results and Discussion section, but it is not the appropriate section.
Author response: Now moved to the method section.
The construction of a sustainable workforce and environmental sustainability variables is not clear (lines 277-281).
Author response: Detail is provided in the method section see lines from 243-251.
I suggest to clarify the type of reports from which data were extracted, how the content analysis is developed and the software used to do it. The sources of the financial and control variables should be known.
Author response: Clarified now.
The time span adopted (2016-2018), based on three years’ data, is not very wide to verify the effects of the implementation of the sustainable workplace policies on the environmental sustainability and financial performance.
Author response: Detail of this short span provided in the method section.
With reference to the limitations of the study, it could be considered that the results cannot be generalized over the Malaysian context.
Author response: Now included in the limitation section of the study.
Reviewer 2 Report
The paper wants to investigate the relationship between a dimension of social sustainability (employee’s workplace) and environmental sustainability and the link of these two variables with financial performance. The first topic related to integrating environmental goals into human resource management, in general, and into workplace sustainability, in particular, is new and very interesting. However, I suggest rejecting the paper as the authors don’t appear to deal with this topic clearly. In my opinion, the reader gets confused about the relationship between workplace sustainability and environmental sustainability. Not all workplace sustainability practices reported in Appendix 1 improve environmental sustainability, but only those integrating environmental goals. If employees are selected on green criteria or if their remuneration is linked to environmental goals, the authors can investigate whether workplace sustainability influences environmental sustainability. If not, the relationship between the two dimensions is not clear.
Concerning methodology, I think that there is room for improvement. The authors justify the choice of Malaysia as the country for their analysis properly, even if, in my opinion, they could describe the regulatory steps concerning the attention of the Malaysian government to sustainability more precisely. On the other hand, in the same way, they should justify the choice of their analysis period (2016-2018). In addition, it is not clear how the authors select sample firms and, above all, why they didn’t consider all Malaysian public listed firms.
One of the main variables is financial performance that can be proxied by both market variables and accounting variables. The authors neglect to discuss these alternatives and, at the end, they choose an accounting proxy while their sample is composed of listed firms without justifying their choice.
With panel data, the use of the P-OLS estimator doesn’t seem completely correct. Using the fixed-effects estimator could be better. On the other hand, if the authors use the P-OLS estimator and include year and industry dummies because they want to analyze industry effects, they should describe their methodological choices more clearly.
I suggest using the natural logarithm for age and thinking if it is possible that the mean ROE can be greater than its maximum value.
Minor worries
1) I don’t understand why the authors select “Corporate Social Responsibility Disclosure”, “Islamic banking industry” and “AAOIFI” as keywords. The focus of the paper seems neither Islamic world nor disclosure of Corporate Social Responsibility.
2) I don’t understand why some references are reported in the text using the first author's surname, while others are reported using their number in the References list.
3) Not only sustainability data, but all data can generate endogeneity problems due to omitted variables bias, measurement error, and reverse causality (line 307-309).
4) I suggest writing the line 296-297 better.
Author Response
The paper wants to investigate the relationship between a dimension of social sustainability (employee’s workplace) and environmental sustainability and the link of these two variables with financial performance. The first topic related to integrating environmental goals into human resource management, in general, and into workplace sustainability, in particular, is new and very interesting. However, I suggest rejecting the paper as the authors don’t appear to deal with this topic clearly. In my opinion, the reader gets confused about the relationship between workplace sustainability and environmental sustainability. Not all workplace sustainability practices reported in Appendix 1 improve environmental sustainability, but only those integrating environmental goals. If employees are selected on green criteria or if their remuneration is linked to environmental goals, the authors can investigate whether workplace sustainability influences environmental sustainability. If not, the relationship between the two dimensions is not clear.
Author Response: Thank you very much for the comment, however, the detail is already provided in the introduction, LR and hypotheses development sections of the study.
Concerning methodology, I think that there is room for improvement. The authors justify the choice of Malaysia as the country for their analysis properly, even if, in my opinion, they could describe the regulatory steps concerning the attention of the Malaysian government to sustainability more precisely. On the other hand, in the same way, they should justify the choice of their analysis period (2016-2018). In addition, it is not clear how the authors select sample firms and, above all, why they didn’t consider all Malaysian public listed firms.
Author Response: Detail is now provided in the methodology section.
One of the main variables is financial performance that can be proxied by both market variables and accounting variables. The authors neglect to discuss these alternatives and, at the end, they choose an accounting proxy while their sample is composed of listed firms without justifying their choice.
Author Response: Complete detail is now provided in the population and sampling sections.
With panel data, the use of the P-OLS estimator doesn’t seem completely correct. Using the fixed-effects estimator could be better. On the other hand, if the authors use the P-OLS estimator and include year and industry dummies because they want to analyze industry effects, they should describe their methodological choices more clearly.
Author Response: The model was suffering from the problem of endogeneity and the previous studies argued that in the case of endogeneity the results of the GLS random and fixed models are biased. Hence, the study applied OLS and 2SLS technique which is highly recommended by the extant literature.
I suggest using the natural logarithm for age and thinking if it is possible that the mean ROE can be greater than its maximum value.
Author Response: The proxy of firm age was adopted from the previous authors as it is. We tested the mean values and it is the same as before.
Minor worries
1) I don’t understand why the authors select “Corporate Social Responsibility Disclosure”, “Islamic banking industry” and “AAOIFI” as keywords. The focus of the paper seems neither Islamic world nor disclosure of Corporate Social Responsibility.
Author Response: It was erroneously left from the previous format of MDPI papers, however, now corrected.
2) I don’t understand why some references are reported in the text using the first author's surname, while others are reported using their number in the References list.
Author Response: Now corrected.
3) Not only sustainability data, but all data can generate endogeneity problems due to omitted variables bias, measurement error, and reverse causality (line 307-309).
Author Response: Most of the previous studies assumed sustainability as endogenous, hence, followed the previous authors and consider sustainability and endogenous.
4) I suggest writing the line 296-297 better.
Author Response: Rewrite accordingly.
Reviewer 3 Report
Dear authors
Doubtful keywords to choose. It is not entirely clear why Islamic 41 banking industry of Pakistan was chosen, that for analysis only Malaysia was chosen, and financial institutions just 9 from 100 analyzed organizations. In abstract written more about sustainability, but keywords-corporate social responsibility.
In the whole paper there is a double feeling: on one hand it is big work and analysis done, but on another- mixed main terminology in the theoretical part. In my opinion there are two different subjects with different criterias and ways how to measure them: corporate social responsibility and sustainable development. Authors very flexible use these terms. But for analysis authors returned to GRI. So in my opinion authors should write only on CSR and not to write in abstract and introduction about sustainable development at all. I didn't find an answer to the research question "what is the level of workplace sustainability practices". Please reformulate research questions or write answers. Also needed to avoid self-citations of yourself.
sincerely
Author Response
Doubtful keywords to choose. It is not entirely clear why Islamic 41 banking industry of Pakistan was chosen, that for analysis only Malaysia was chosen, and financial institutions just 9 from 100 analyzed organizations. In abstract written more about sustainability, but keywords-corporate social responsibility.
Authors response: It was erroneously left from the previous format of MDPI papers, however, now corrected.
In the whole paper there is a double feeling: on one hand it is big work and analysis done, but on another- mixed main terminology in the theoretical part. In my opinion there are two different subjects with different criterias and ways how to measure them: corporate social responsibility and sustainable development. Authors very flexible use these terms. But for analysis authors returned to GRI. So in my opinion authors should write only on CSR and not to write in abstract and introduction about sustainable development at all. I didn't find an answer to the research question "what is the level of workplace sustainability practices". Please reformulate research questions or write answers. Also needed to avoid self-citations of yourself.
Authors response: Keywords and research questions are now corrected.
Round 2
Reviewer 1 Report
- After the reduction of the abstract, it summarizes yet all the content of the manuscript.
- Although the SDGs (No. 8, 13 and 17) are not discussed in a deepen way in the introduction section, they are well-specified in the abstract.
- Now, the new section 3.2 the statistical techniques embraces the techniques involved to test the hypothesis (OLS and 2SLS).
- The construction process of the sustainable workforce and environmental sustainability variables is clearer than the previous version of the manuscript.
- After the revision, the current version of the paper embraces the sources of the data about the workplace and environmental sustainability practices and control variables.
- The brief time span adopted (2016-2018) to conduct the analysis, continues to be a limitation.
- “Fourthly, some other developing countries should also be considered in future studies for the generalizability of the results.” The limitation added in the text is in line with our suggestion.
Author Response
All the above comments are addressed as the reviewer show satisfaction.
Reviewer 2 Report
I had suggested rejecting the paper as, in my opinion, the paper's quality was not suitable for major revisions during my first revision. Despite this, as required from the journal, I re-read the paper and focused on all the changes. Although I recognize the authors' huge effort and commitment to improving the paper's quality, I confirm my recommendation of rejecting the paper as the main critical points have not been solved yet.
First, the relationship between workplace sustainability and environmental sustainability remains unclear even if the authors attempt to give more details. More precisely, as reported in Appendix 1, the authors don’t consider the specific aspects of workplace sustainability linked to environmental sustainability, but they also consider the main traditional dimensions of the first concept (diversity and equal opportunities, ethical values etc.). Moreover, the relationship between workplace sustainability and environmental sustainability is described only thanks to previous literature and not thanks to a specific theory. The authors indeed cite stakeholder theory, but to emphasize the firm's focus on the multiple stakeholders' demand without clarifying the details of the relationship between workplace sustainability and environmental sustainability.
Second, the authors don’t explain the analysis period's selection and the choice of ROE as a proxy for financial performance. The mean value of ROE is equal to 9.09, still much higher than the maximum value, equal to 0.89.
Third, the authors continue to use age and not the corresponding natural logarithm. In the last version, the text from line 259 to line 263 appears not correctly formatted.
Author Response
Comment 1
I had suggested rejecting the paper as, in my opinion, the paper's quality was not suitable for major revisions during my first revision. Despite this, as required from the journal, I re-read the paper and focused on all the changes. Although I recognize the authors' huge effort and commitment to improving the paper's quality, I confirm my recommendation of rejecting the paper as the main critical points have not been solved yet.
Authors’ Response
We always appreciate the critical comments of the reviewers, however, the current case of the workplace and environmental sustainability is novel, and very few studies are available in the relationship. Hence, the study focuses to develop the link between the workplace and environmental sustainability, and financial performance. Moreover, the study also focuses to address the methodological issues in the sustainability literature and overcome the problem of endogeneity. Therefore, in the end, it is suggested that this study will add new insight into the aforementioned relationship and would be beneficial for future studies in the area concerned.
Comment 2
First, the relationship between workplace sustainability and environmental sustainability remains unclear even if the authors attempt to give more details. More precisely, as reported in Appendix 1, the authors don’t consider the specific aspects of workplace sustainability linked to environmental sustainability, but they also consider the main traditional dimensions of the first concept (diversity and equal opportunities, ethical values etc.).
Authors’ Response
Further detail is also provided about the relationship between the workplace and environmental sustainability. As far as the data collection index of the workplace and environmental sustainability is concerned, the study adopted from the Global Reporting Initiative (GRI) framework which is one of the prominent recommendations of the United Nation Environmental Program (UNEP), references are already provided in the manuscript.
Comment 3
Moreover, the relationship between workplace sustainability and environmental sustainability is described only thanks to previous literature and not thanks to a specific theory. The authors indeed cite stakeholder theory, but to emphasize the firm's focus on the multiple stakeholders' demand without clarifying the details of the relationship between workplace sustainability and environmental sustainability.
Authors’ Response
Further detail is now provided in both theory and the subject relationship see section 2.3.1.
Comment 4
Second, the authors don’t explain the analysis period's selection and the choice of ROE as a proxy for financial performance. The mean value of ROE is equal to 9.09, still much higher than the maximum value, equal to 0.89.
Authors’ Response
The analysis period is now explained in complete detail. The proxy of ROE was adopted from the previous literature and now the references and detail are provided. Lastly, the mean value of ROE was written erroneously as 9.09 instead of 0.09, hence, corrected accordingly.
Comment 5
Third, the authors continue to use age and not the corresponding natural logarithm. In the last version, the text from line 259 to line 263 appears not correctly formatted.
Authors’ Response
The proxy of the age was adopted from the previous literature, references are already provided.
Round 3
Reviewer 2 Report
I can tell that the authors accepted my suggestions only partly, even if I noticed that they considered all my comments. For example, they revised the parts related to age and ROE. However, the main problems remain: they still don’t use the natural logarithm for age and don’t justify because they approximate the financial performance of listed firms using ROE.
They don’t justify the period of analysis precisely. For example, they don’t indicate the year of the introduction of the Malaysian Code of Corporate Governance (MCCG). Maybe, this information could help the reader to understand the choice of the period of analysis.
However, in my opinion, the main problem is the lack of a theoretical framework. The authors cite the Stakeholder Theory to justify a higher perspective of the firms that have to be interested not only in shareholders but also in the bigger category of stakeholders. However, the reader doesn’t understand how “Workplace sustainability integrates firms’ environmental objectives in the human resource functions like the recruitment, training, and reviewing the performance and remuneration of the employees to assist firms in ensuring carbon emission and earning carbon credits”. In my opinion, this should be the focus of the paper. However, at the end, I think that the paper can be published if the editor considers the issues I highlight minor details.
Author Response
Dear Reviewer 2, thank you very much for your valuable comments on the subject paper. Please find the review of your comments as follows.
Comment 1
I can tell that the authors accepted my suggestions only partly, even if I noticed that they considered all my comments. For example, they revised the parts related to age and ROE. However, the main problems remain: they still don’t use the natural logarithm for age and don’t justify because they approximate the financial performance of listed firms using ROE.
Authors Reply
As mentioned in the earlier review that most of the previous authors used the same proxy of firm age as mentioned in the following papers to name a few. Hence, the authors rely on the same proxy of firm age.
- Dissanayake, D.; Tilt, C.; Xydias-Lobo, M. Sustainability reporting by publicly listed companies in Sri Lanka. J. Clean. Prod. 2016, 129, 169–182, doi:10.1016/j.jclepro.2016.04.086.
- Zahid, M.; Rahman, H.U.; Muneer, S.; Butt, B.Z.; Isah-Chikaji, A.; Memon, M.A. Nexus Between Government Initiatives, Integrated Strategies, Internal Factors and Corporate Sustainability Practices in Malaysia. J. Clean. Prod. 2019, 241, 118329, doi:10.1016/j.jclepro.2019.118329.
- Rehman, Z.U.; Zahid, M.; Rahman, H.U.; Asif, M.; Alharthi, M.; Irfan, M.; Glowacz, A. Do Corporate Social Responsibility Disclosures Improve Financial Performance? A Perspective of the Islamic Banking Industry in Pakistan. Sustainability 2020, 12, 1–18, doi:10.3390/su12083302.
- Zahid, M.; Rahman, H.U.; Khan, M.; Ali, W.; Shad, F. Addressing endogeneity by proposing novel instrumental variables in the nexus of sustainability reporting and firm financial performance : A step-by-step procedure for non- experts. Bus. Strateg. Environ. 2020, 29, 3086–3103, doi:10.1002/bse.2559.
Comment 2
They don’t justify the period of analysis precisely. For example, they don’t indicate the year of the introduction of the Malaysian Code of Corporate Governance (MCCG). Maybe, this information could help the reader to understand the choice of the period of analysis.
Authors Reply
The reference year is now provided.
Comment 3
However, in my opinion, the main problem is the lack of a theoretical framework. The authors cite the Stakeholder Theory to justify a higher perspective of the firms that have to be interested not only in shareholders but also in the bigger category of stakeholders. However, the reader doesn’t understand how “Workplace sustainability integrates firms’ environmental objectives in the human resource functions like the recruitment, training, and reviewing the performance and remuneration of the employees to assist firms in ensuring carbon emission and earning carbon credits”. In my opinion, this should be the focus of the paper. However, at the end, I think that the paper can be published if the editor considers the issues I highlight minor details.
Authors Reply
The same theoretical explanation is provided in the theoretical framework section of the manuscript please see line 157 to 168 section 2.3.1.
Thank you.